# Generative network modeling reveals quantitative definitions of bilateral symmetry exhibited by a whole insect brain connectome

Benjamin D Pedigo[1]*, Mike Powell[1†], Eric W Bridgeford[2], Michael Winding[3,4,5], Carey E Priebe[6], Joshua T Vogelstein[1]

[1]Department of Biomedical Engineering, Johns Hopkins University, Baltimore, United States; [2]Department of Biostatistics, Johns Hopkins University, Baltimore, United States; [3]Department of Zoology, University of Cambridge, Cambridge, United Kingdom; [4]Neurobiology Division, MRC Laboratory of Molecular Biology, Cambridge, United Kingdom; [5]Janelia Research Campus, Howard Hughes Medical Institute, Ashburn, United States; [6]Department of Applied Mathematics and Statistics, Johns Hopkins University, Baltimore, United States

*For correspondence:
bpedigo@jhu.edu

Present address: [†]Department of Mathematical Sciences, United States Military Academy, New York, United States

Competing interest: The authors declare that no competing interests exist.

**Abstract** Comparing connectomes can help explain how neural connectivity is related to genetics, disease, development, learning, and behavior. However, making statistical inferences about the significance and nature of differences between two networks is an open problem, and such analysis has not been extensively applied to nanoscale connectomes. Here, we investigate this problem via a case study on the bilateral symmetry of a larval *Drosophila* brain connectome. We translate notions of 'bilateral symmetry' to generative models of the network structure of the left and right hemispheres, allowing us to test and refine our understanding of symmetry. We find significant differences in connection probabilities both across the entire left and right networks and between specific cell types. By rescaling connection probabilities or removing certain edges based on weight, we also present adjusted definitions of bilateral symmetry exhibited by this connectome. This work shows how statistical inferences from networks can inform the study of connectomes, facilitating future comparisons of neural structures.

## Editor's evaluation

This important work demonstrates a significant asymmetry between the connectivity statistics of the left and right hemispheres of the *Drosophila* larva brain. The evidence supporting the conclusions is compelling and represents a first step toward the development of statistical tests for comparing pairs of connectomes more generally. This work will therefore be of interest to the broad neuroscience community.

## Introduction

Connectomes – maps of neural wiring – have become increasingly important in neuroscience, and are thought to be an important window into studying how connectivity relates to neural activity, evolution, disease, genetics, and learning (*Vogelstein et al., 2019*; *Abbott et al., 2020*; *Barsotti et al., 2021*; *Galili et al., 2022*). However, many of these pursuits in connectomics depend on being able to compare networks. For instance, to understand how memory relates to connectivity, one would need

to map a connectome which has learned something and one which has not, and then assess whether and how the two networks are different. To understand how a gene affects connectivity, one would need to map a connectome from an organism with a genetic mutation and one from a wild-type organism, and then assess whether and how the two networks are different. Authors have advocated for comparing connectomes across the phylogenetic tree of life (*Barsotti et al., 2021*; *Galili et al., 2022*), disease states (*Abbott et al., 2020*), life experiences (*Galili et al., 2022*; *Abbott et al., 2020*), development (*Galili et al., 2022*), and sex (*Galili et al., 2022*).

Several recent works have already started toward this goal of comparative connectomics. *Gerhard et al., 2017*, compared the connections in the nerve cord (the insect equivalent of a spinal cord) of the L1 and L3 stages of the larval *Drosophila melanogaster* to understand how these connections change as the animal develops. Similarly, *Witvliet et al., 2021*, collected connectomes from *Caenorhabditis elegans* at various life stages, and examined which connections were stable and which were dynamic across development. *Cook et al., 2019*, generated connectomes for both a male and hermaphrodite *C. elegans* worm to understand which aspects of this organism's wiring diagram differ between the sexes. *Valdes-Aleman et al., 2021*, made genetic perturbations to different individual *D. melanogaster* fly larva, and examined how these perturbations affected the connectivity of a local circuit in the organism's nerve cord. Viewed through the lens of the wiring diagrams alone (i.e. ignoring morphology, subcellular structures, etc.), these pursuits all amount to comparing two or more networks.

In addition to those described above, one comparison that has been prevalent in the connectomics literature is to assess the degree of left/right structural similarity of a nervous system. *Bilateria* is a group of animals which have a left/right structural symmetry. This clade is thought to have emerged around 550 million years ago (*Fedonkin and Waggoner, 1997*), making it one of the oldest groups of animals. Most organisms studied in neuroscience (including *C. elegans*, *D. melanogaster*, mice, rats, monkeys, and humans) are all bilaterians. While functional asymmetries in the brain have been discovered, this axis of structural symmetry is generally thought to extend to the brain (*Hobert, 2014*).

Connectomic studies have investigated this structural similarity in various ways. The degree of left/right symmetry in a single connectome has often been studied as a proxy or lower bound for the amount of stereotypy that one could expect between connectomes of different individuals. *Lu et al., 2009*, reconstructed the connectome of the axons projecting to the interscutularis muscle on the left and right sides of two individual mice. They found that the branching patterns of axons between the left and right sides within one animal were no more similar than a comparison between the two animals, and also no more similar than two random branching patterns generated by a null model. In contrast, *Schlegel et al., 2021*, found a striking similarity between the morphologies of neurons (as measured by NBLAST; *Costa et al., 2016*) in the left and right hemispheres of the *D. melanogaster* antennal lobe, and a similar level of stereotypy between the antennal lobes of two different individuals. *Cook et al., 2019*, used the observed level of left-right variability in a *C. elegans* hermaphrodite connectome as a proxy for the amount of variability in connectivity between individuals, assuming that one should expect the connectomes of the left and right to be the same up to developmental and experiential variability. Conversely, they also point out the fact that the ASEL neuron (on the left side) projects more strongly to neuron class AWC than the analogous version on the right, verifying this difference via fluorescent labeling in another animal. Similarly, in confocal imaging studies in *Drosophila*, the vast majority of genetically defined cell types were found to have bilaterally symmetric morphologies, but with one notable exception. A population of neurons projecting to the aptly named asymmetric body were found to preferentially target this structure on the right hemisphere in most animals (*Jenett et al., 2012*; *Wolff and Rubin, 2018*), and this bias was even found to be related to function (*Pascual et al., 2004*). These studies highlight the complicated relationship between neuroscientists and bilateral symmetry: at times, we may assume that the left and right sides of a nervous system are in some sense the same in expectation, but at other times we find marked, reproducible differences between them. To date, no study (to our knowledge) has framed this question of bilateral symmetry of connectivity as a statistical hypothesis comparing two networks.

In this work, we compare the connectivity of the left and the right hemispheres of an insect connectome from the perspective of statistical hypothesis testing. Motivated by the discussion above, in this work we make three major contributions: (1) we formally state several notions of bilateral symmetry for connectomes as statistical hypotheses, (2) we present test procedures for each of these hypotheses

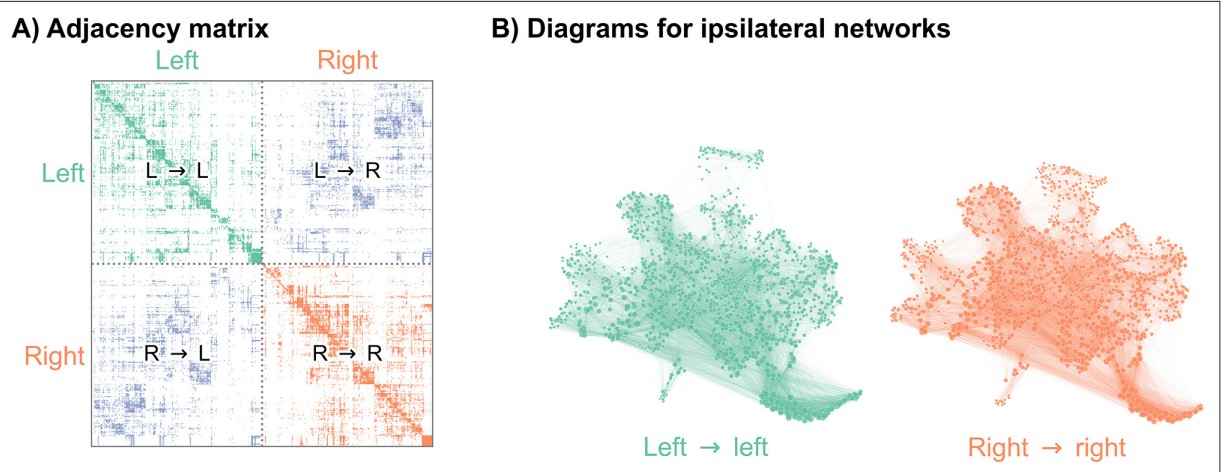

**Figure 1.** Visualizations of a larval *Drosophila* brain connectome from **Winding et al., 2023**. (**A**) Adjacency matrix for the full brain connectome network, sorted by brain hemisphere. Note that we ignore the left → right and right → left (contralateral) subgraphs in this work. (**B**) Network layouts for the left → left and right → right subgraphs.

The online version of this article includes the following source data for figure 1:

**Source data 1.** *Drosophila* larva brain connectome.

of bilateral symmetry, and (3) we demonstrate the utility of these tests for understanding the significance and nature of bilateral symmetry/asymmetry in the brain of a *D. melanogaster* larva. In doing so, we provide a framework and methodology for any neuroscientist wishing to compare two networks, facilitating future work in comparative connectomics. We also provide Python implementations and documentation for the statistical tests for network comparison developed in this work.

## Results

### Connectome of a larval *Drosophila* brain

Recently, authors mapped a connectome of the brain of a *D. melanogaster* larva (**Winding et al., 2023**). To understand how the neurons in this brain were connected to each other, the authors first imaged this brain using electron microscopy (**Ohyama et al., 2015**), and then manually reconstructed each neuron and its pre- and post-synaptic contacts. This synaptic wiring diagram consists of 3016 neurons and over 548,000 synapses. We represent this connectome as a network, with nodes representing neurons and edges representing some number of synapses between them (**Figure 1**). Importantly, this work yielded a complete reconstruction of both the left and right hemispheres of the brain. In order to assess bilateral symmetry, we focused on the left-to-left and right-to-right (ipsilateral) induced subgraphs. While there are conceivable ways to define bilateral symmetry which include the contralateral connections, we did not consider them here in order to restrict our methods to the more widely applicable case of two-network-sample testing. More details on how we created the networks to compare here are available in Network construction. This process yielded a 1506 neuron network for the left hemisphere, and a 1506 neuron network for the right (note that the number of nodes in the two hemispheres need not have been exactly the same).

We sought to understand whether these two networks were significantly different according to some definition, in order to characterize whether this brain was bilaterally symmetric. As with any statistical hypothesis test, this required that we make some modeling assumptions about the nature of the networks being compared. We stress that our subsequent results should be interpreted in light of these models and what they do (and do not) tell us about these networks (see **Váša and Mišić, 2022**, for an excellent discussion of this point in network neuroscience, and see Limitations for a discussion of alternative modeling assumptions). For all of our models, we treated the networks as directed (since we knew the direction of synapses), unweighted (creating an edge when there was one or more synapse between neurons unless otherwise specified), and loopless (since we ignored any observed self-loops). We made no assumptions about whether individual neurons in the left hemisphere

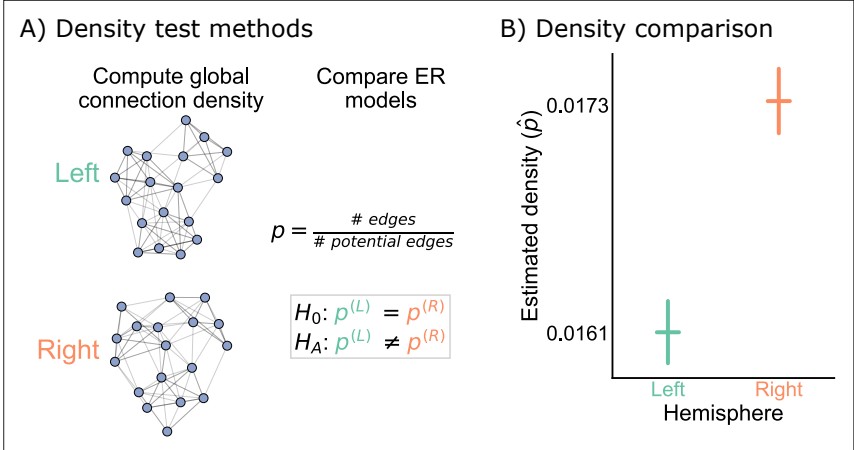

**Figure 2.** Comparison of left and right hemisphere networks via the density test. (**A**) Diagram of the methods used for testing based on the network density. See Erdos-Renyi model and density testing for more details. (**B**) The estimated density $\hat{p}$ (probability of any edge averaged across the entire network) for the left hemisphere is ~0.016, while for the right it is ~0.017 – this makes the left density ~0.93 that of the right. Vertical lines denote 95% confidence intervals for this estimated parameter $\hat{p}$. The p-value for testing the null hypothesis that these densities are the same is $<10^{-23}$ (two-sided chi-squared test), meaning very strong evidence to reject the null that the left and right hemispheres have the same density.

correspond with individual neurons in the right hemisphere. Next, we detail a series of more specific models, what aspects of the networks they characterize, and how we construct a hypothesis test from each.

## Density test

Our first test of bilateral symmetry was based on perhaps the simplest network model, the Erdos-Renyi (ER) model (*Gilbert, 1959*; *Erdős and Rényi, 1960*), which models each potential edge as independently generated with the same probability, $p$. Comparing two networks under the ER model amounts to simply comparing their densities (*Figure 2A*).

$$H_0 : p^{(L)} = p^{(R)} \text{ vs. } H_A : p^{(L)} \neq p^{(R)} \tag{1}$$

This comparison of probabilities can be tested using well-established statistical machinery for two-sample tests under the binomial distribution (see ER model and density testing for more details). We refer to this procedure as the density test.

*Figure 2B* shows the comparison of the network densities between the left and right hemisphere networks. The densities of the left and right are ~0.016 and ~0.017, respectively, making the density of the left ~0.93 that of the right. To determine whether this is a difference likely to be observed by chance under the ER model, we ran a two-sided chi-squared test, which tests whether the probabilities of two independent binomials are significantly different. This test yielded a p-value $<10^{-23}$, suggesting that we have strong evidence to reject this version of our hypothesis of bilateral symmetry. While the ratio of the estimated densities is only ~0.93, this extremely small p-value resulted from the large sample size for this comparison, as there are 2,266,530 potential edges on both the left and the right sides.

To our knowledge, when neuroscientists have considered the question of bilateral symmetry, they have not meant such a simple comparison of network densities. In many ways, the ER model is too simple to be an interesting description of connectome structure. However, it is also striking that perhaps the simplest network comparison produced a significant difference between brain hemispheres for this brain. It is unclear whether this difference in densities is biological (e.g. a result of slightly differing rates of development for this individual), an artifact of how the data was collected (e.g. technological limitations causing slightly lower reconstruction rates on the left hemisphere), or something else entirely. Still, in addition to highlighting a simple departure from symmetry in this dataset, the density test result also provides important considerations for other tests. More complicated models

**Table 1.** Neuron group definitions, categorizations from *Winding et al., 2023*.

| Acronym | Full name |
|---|---|
| Ascending | Ascending neurons from the ventral nerve cord |
| CN | Convergence neurons, receiving input from lateral horn and mushroom body |
| $DN^{SEZ}$ | Descending neurons (to the sub-esophageal zone) |
| $DN^{VNC}$ | Descending neurons (to the ventral nerve cord) |
| KC | Kenyon cells |
| LHN | Lateral horn neurons |
| LN | Local neurons |
| MB-FBN | Mushroom body feedback neurons |
| MB-FFN | Mushroom body feedforward neurons |
| MBIN | Mushroom body input neurons |
| MBON | Mushroom body output neurons |
| Other | Neurons lacking any other categorization |
| PN | Projection neurons |
| $PN^{Somato}$ | Somatosensory projection neurons |
| Pre-$DN^{SEZ}$ | Neurons projecting to $DN^{SEZ}$s |
| Pre-$DN^{VNC}$ | Neurons projecting to $DN^{VNC}$s |
| RGN | Ring gland neurons |
| Sensory | Sensory neurons |

of symmetry could compare other network statistics – say, the clustering coefficients, the number of triangles, and so on. These statistics, as well as the model-based parameters we will consider in this paper, are strongly related to the network density (*Suarez et al., 2022*; *Chen et al., 2021*). Thus, if the densities are different, it is likely that tests based on any of these other test statistics will also reject the null hypothesis of bilateral symmetry. Later, we describe methods for adjusting for a difference in density in other tests for bilateral symmetry.

## Group connection test

To understand whether this broad difference between the hemispheres can be localized to a specific set of connections, we next tested bilateral symmetry by making an assumption that the left and right hemispheres both come from a stochastic block model (SBM). Under the SBM, each neuron is assigned to a group, and the probability of any potential edge is a function of the groups to which the source and target neurons belong. For instance, the probability of a connection from a neuron in group $k$ to a neuron in group $l$ is set by the parameter $B_{kl}$, where $B$ is a $K \times K$ matrix of connection probabilities if there are $K$ groups. Here, we used broad cell type categorizations from *Winding et al., 2023*, to determine each neuron's group (see *Figure 3—figure supplement 1* for the number of neurons in each group in each hemisphere, see *Table 1* for naming conventions). Alternatively, there are many methods for estimating these assignments to groups for each neuron which we do not explore here (see Limitations for discussion on this point). Under the SBM with a fixed group assignment for each node, testing for bilateral symmetry amounts to testing whether the group-to-group connection probability matrices, $B^{(L)}$ and $B^{(R)}$, are the same.

$$H_0 : B^{(L)} = B^{(R)} \text{ vs. } H_A : B^{(L)} \neq B^{(R)} \tag{2}$$

Rather than having to compare one probability as in *Equation 1*, we were interested in comparing all $K^2$ group-to-group connection probabilities between the SBM models for the left and right hemispheres. We developed a novel statistical hypothesis test for this comparison, which uses many tests to compare each of the group-to-group connection probabilities, followed by appropriate correction

for multiple comparisons (when examining the individual group-to-group connections) or combination of p-values (when assessing the overall null hypothesis in *Equation 2*). Details on the methodology used here is provided in SBM and group connection testing, and is shown as a schematic in *Figure 3A*. We refer to this procedure as the group connection test.

*Figure 3B* shows both of the estimated group-to-group probability matrices, $\hat{B}^{(L)}$ and $\hat{B}^{(R)}$. From a visual comparison of $\hat{B}^{(L)}$ and $\hat{B}^{(R)}$, the group-to-group connection probabilities appear qualitatively similar. Note also that some group-to-group connection probabilities are zero, making it nonsensical to do a comparison of probabilities. We highlight these elements in the matrices with explicit '0's, and note that we did not run the corresponding test in these cases. *Figure 3C* shows the p-values from all 285 tests that were run to compare each element of these two matrices. After multiple comparisons correction, seven tests produced p-values less than $\alpha = 0.05$, indicating that we could reject the null hypothesis that those specific connection probabilities are the same between the two hemispheres. We also combined all (uncorrected) p-values, yielding an overall p-value for the entire null hypothesis (*Equation 2*) of equivalence of group-to-group connection probabilities of $<10^{-7}$.

Taken together, these results suggest that while the group-to-group connections are roughly similar between the two hemispheres, they are not the same under this model. Notably, there are seven group-to-group connections which were significantly different: Kenyon cells (KC) → KC, lateral horn neurons (LHN) → other, other → LHN, other → other, projection neurons (PN) → LHN, somatosensory projection neurons ($PN^{Somato}$) →other, and $PN^{Somato}$ $PN^{Somato}$. We stress that, as with any statistical test, a lack of a significant difference (e.g. in other subgraphs) could be the result of the null hypothesis of no difference being true, or simply from a lack of power against a particular alternative (see *Figure 3—figure supplement 2* and *Figure 3—figure supplement 3* for analysis of the power of this test in simulation, and *Helwegen et al., 2023*, for an excellent discussion on this point). Nevertheless, knowing some neuron groups which are wired significantly differently between the two hemispheres highlights the interpretability of this test. If a neuroscientist wanted to study mechanisms which could cause bilateral asymmetries in the brain, these seven group-to-group connections would be prime candidates for investigation.

However, in Density test, we saw that the densities of the two networks are significantly different. $p$, the density of the network, can be thought of as a weighted average of the individual group-to-group connection probabilities, $B$. Should we then be surprised that if the density is different, the group-to-group connection probabilities are, too? Interestingly, for all the group-to-group connection probabilities which are different, the probability on the right hemisphere (which has the greater density) is higher (*Figure 3D*). We consider this phenomenon in the next section.

## Density-adjusted group connection test

Next, we examined whether the group-to-group connection probabilities on the right are simply a 'scaled-up' version of those on the left. *Figure 3D* showed that for all the individual connections which are significant, the connection probability on the right hemisphere is higher. This is consistent with the hypothesis stated above, which predicts that the connection probabilities in $B^{(R)}$ should be consistently higher than those in $B^{(L)}$.

We thus created a test for this notion of bilateral symmetry in group-to-group connections (up to a density adjustment):

$$H_0 : B^{(L)} = cB^{(R)} \text{ vs. } H_A : B^{(L)} \neq cB^{(R)} \tag{3}$$

Note that these adjusted hypothesis do not test whether the density across all subgraphs of the left or right hemisphere networks are the same; rather, they are asking wither a single scaling factor ($c$ in *Equation 3*) makes any significant density *differences* disappear from our previous comparison. To implement this hypothesis test, we first computed the density correcting constant $c$, which is simply the ratio of the left to the right hemisphere densities, finding that $c \approx 0.93$. Then, we replaced each of the component tests in the group connection test with a modified version of the standard chi-squared test for non-unity probability ratios (see Density-adjusted group connection testing for more details) (*Miettinen and Nurminen, 1985*). We refer to this procedure as the density-adjusted group connection test (*Figure 4A*). The p-values for each of the component tests for the density-adjusted group connection test are shown in *Figure 4B*. After correction for multiple comparisons, there are two group-to-group connections which are significantly different (at significance level 0.05): KC →

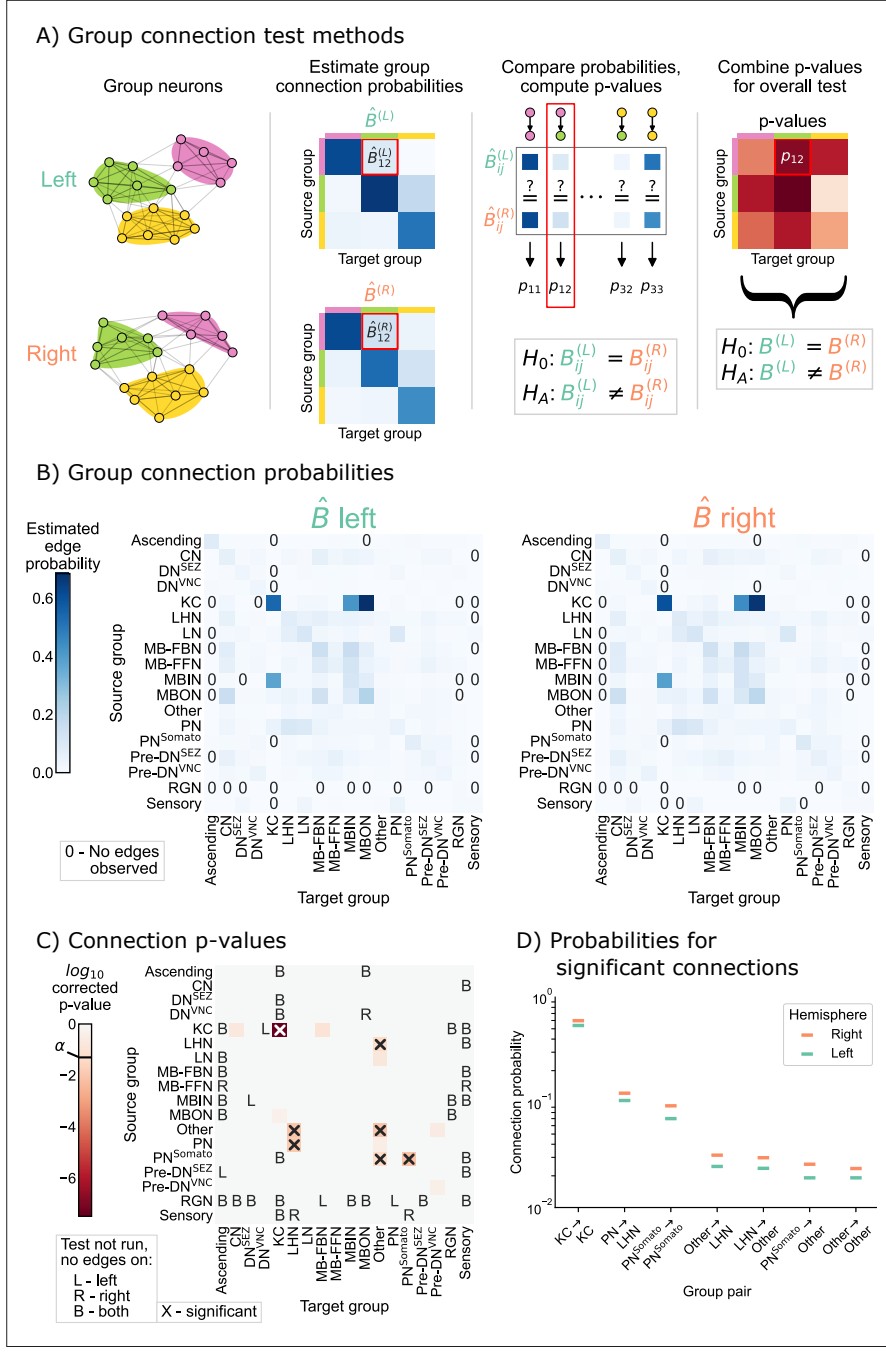

**Figure 3.** Comparison of left and right hemisphere networks via the group connection test. (**A**) Description of methodology for the group connection test. See SBMs and group connection testing for more details. (**B**) Estimated group-to-group connection probabilities for both hemispheres. Note that they appear qualitatively similar. Estimated probabilities which are zero (no edge was present between that pair of groups) are indicated with a '0' in those cells. (**C**) p-Values (after multiple comparisons correction) for each hypothesis test between individual elements of the connection probability matrices. Each cell represents a test for whether a specific group-to-group connection probability is the same on the left and right sides. 'X' denotes a significant p-value after multiple comparisons correction, with significance level $\alpha = 0.05$. 'B' indicates that a test was not run since the estimated probability was zero on both hemispheres, 'L' indicates this was the case on the left only, and 'R' that it was the case on the right only. The individual (uncorrected) p-values were combined using Tippett's method, resulting in an overall p-value (for the null hypothesis that the two group connection probability *matrices* are the same) of $<10^{-7}$. (**D**) Comparison of estimated group-to-group connection probabilities for the group pairs that are significantly different. In each case, the connection probability on the right hemisphere is higher.

*Figure 3 continued on next page*

*Figure 3 continued*

The online version of this article includes the following figure supplement(s) for figure 3:

**Figure supplement 1.** The number of neurons in each neuron group in the left and right hemispheres.

**Figure supplement 2.** Empirical power (in simulations) for tests comparing subgraph connection probabilities (chi-squared test, i.e. each component test in the group connection test).

**Figure supplement 3.** Estimated empirical power for each component of the group connection test in simulations based on the observed *Drosophila* larva brain connectome.

**Figure supplement 4.** Demonstration that the group connection test is both valid and powerful against a range of alternatives in a synthetic simulation based on the observed data.

**Figure supplement 5.** p-Values from the group connection test (as described in *Figure 3*), but using Fisher's exact test for each individual subgraph comparison.

---

convergence neurons (CN) and KC → mushroom body output neurons (MBON). Thus, all significant differences between the hemispheres under this version of the SBM are associated with the Kenyon cells.

## Removing Kenyon cells

Based on the results of *Figure 4C*, we sought to verify that the remaining differences in group-to-group connection probabilities after adjusting for a difference in density can be explained by asymmetry that is isolated to the Kenyon cells. To confirm this, we simply removed the Kenyon cells (i.e. all Kenyon cell nodes and edges to or from those nodes) from both the left and right hemisphere networks, and then re-ran each of the tests for bilateral symmetry presented here (*Figure 5A*). We observed significant differences between the left and right hemispheres for the density and group connection tests when excluding Kenyon cells, yielding p-values of $<10^{-27}$ and $<10^{-2}$, respectively (*Figure 5B and C*). However, for the density-adjusted group connection test, the p-value was ~0.60, indicating that we no longer rejected bilateral symmetry under this definition when the Kenyon cells are excluded from the analysis (*Figure 5D*). This sequence of results suggests that the difference

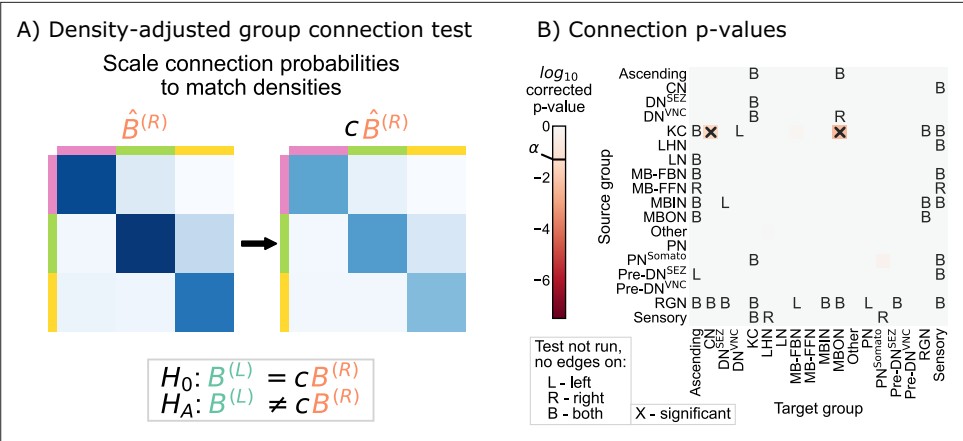

**Figure 4.** Comparison of left and right hemisphere networks via the density-adjusted group connection test. (**A**) Description of methodology for adjusting for a density difference between the two stochastic block models. See SBMs and group connection testing for more details. The adjustment factor (ratio of the left to the right density), c, is ~0.93. (**B**) p-Values for each group-to-group comparison after adjusting for a global density difference. p-Values are shown after correcting for multiple comparisons. Note that there are two significant p-values, and both are in group connections incident to Kenyon cells. These individual (uncorrected) p-values were combined using Tippett's method, resulting in an overall p-value (for the null hypothesis that the two group connection probability *matrices* are the same after correcting for the density difference) of $<10^{-2}$.

The online version of this article includes the following figure supplement(s) for figure 4:

**Figure supplement 1.** Distribution of p-values from experiments in which 2760 edges were randomly removed from the right hemisphere to set the densities of the left and right hemispheres equal, and the group connection test was re-run.

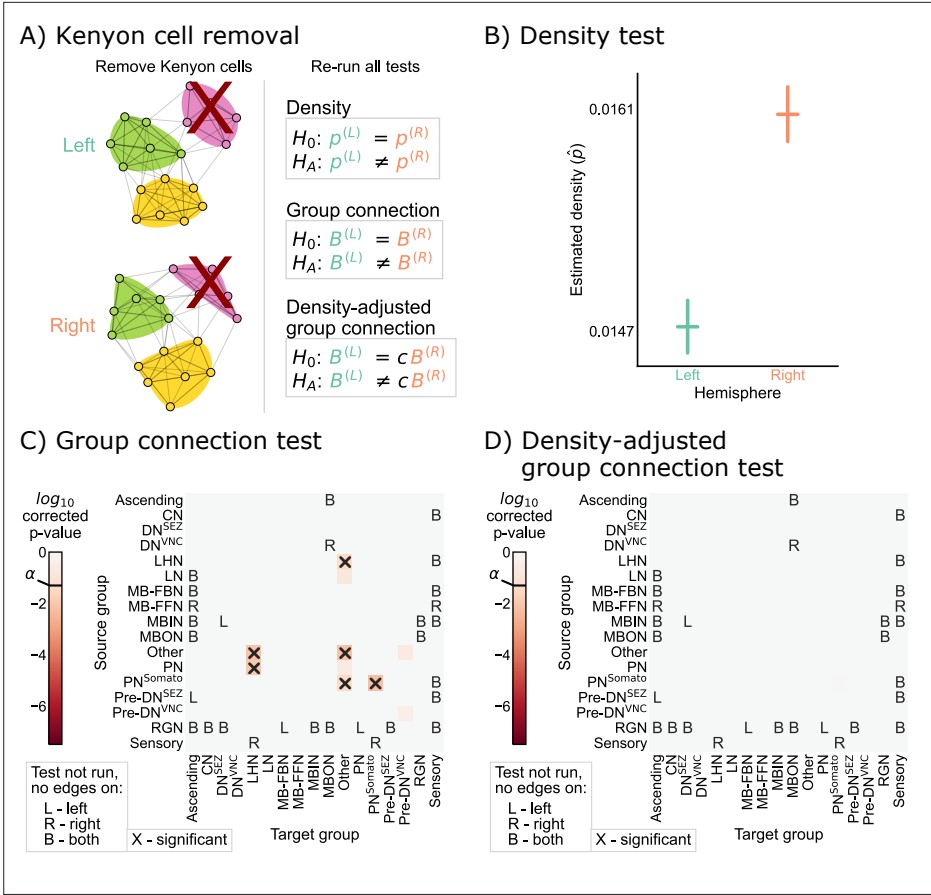

**Figure 5.** Comparison of left and right hemisphere networks when not including Kenyon cells. (**A**) Diagram of the methods used, indicating that Kenyon cells (and any incident edges) were simply removed from the network, and all previously mentioned tests were run again. (**B**) Comparison of network densities, as in *Figure 2B*. The p-value for this comparison is $<10^{-27}$, indicating very strong evidence to reject the null that the two networks share the same density. (**C**) Comparison of group-to-group connection probabilities, as in *Figure 3C*. p-Values are shown for each group-to-group connection comparison (after multiple comparison correction). The (uncorrected) p-values were combined to yield an overall p-value of $<10^{-2}$, showing evidence that the group connection probabilities are not the same even after removing Kenyon cells. (**D**) Comparison of group-to-group connection probabilities after density adjustment, as in *Figure 4C*. p-Values are shown for each group-to-group connection comparison (after multiple comparison correction). Note that there are no longer any significantly different connections. The (uncorrected) p-values were combined to yield an overall p-value of ~0.60. After removing Kenyon cells, there is no longer evidence to reject the null that the group connection probabilities are the same.

between the left and right hemispheres (at least in terms of the high-level network statistics studied here) can be explained as the combination of a global effect (the difference in density) and a cell type-specific effect (the difference in Kenyon cell projection probabilities).

It is noteworthy that the Kenyon cells were the specific cell type where we detected asymmetry after correcting for the density difference. Kenyon cells are involved in associative learning in *Drosophila* and other insects (*Heisenberg, 2003*; *Aso et al., 2014*; *Eichler et al., 2017*). Other studies have suggested that certain connections (specifically from antennal lobe projection neurons to Kenyon cells) are random (*Caron et al., 2013*; *Eichler et al., 2017*). The marked lack of symmetry we observed specifically in the Kenyon cells in the current study could be the result of these features, which suggest their uniquely non-stereotyped patterns of connectivity in this nervous system.

## Edge weight thresholds

Next, we sought to examine how the definition of an edge used to construct our binary network affects the degree of symmetry under each of the definitions considered here. For the networks

considered in the previous sections, we considered an edge to exist if one or more synapses from neurons $i$ to $j$ were in the dataset. To understand how our analysis might change based on this assumption, we considered two types of edge weight threshold schemes for creating a binary network before testing: the first based simply on a threshold on the number of synapses, and the second based on a threshold of the proportion of a downstream neuron's input (*Figure 6A*). By varying the threshold in both schemes, we were able to evaluate many hypotheses about bilateral symmetry, where higher thresholds meant that we only considered the symmetry present in strong edges (*Figure 6B*).

Before running the tests for each of these notions of symmetry, we first examined the distributions of edge weights to get a sense for how we should expect these tests to perform. *Figure 6C and D* displays the distribution (total count) for the synapse count or input proportion edge weights, respectively. The right hemisphere has more connections than the left for all synapse count values (*Figure 6C*), hinting that the density of the right hemisphere will be slightly higher for any potential edge weight threshold using this definition. Conversely, the distributions of weights as an input percentage shows a different trend. For edge weights less than ~1.25%, the right appears to have more edges, but past this threshold, the counts of edges between left and right appear comparable (*Figure 6D*).

*Figure 6E and F* shows the effect of varying these thresholds on the p-values from each of our tests of bilateral symmetry. We observed that for either thresholding scheme (synapse count or input proportion), the p-value for each test generally increased as a function of the threshold – in other words, the left and right hemisphere networks became less significantly different (under the definitions of 'different' we have presented here) as the edge weight threshold increased. Previous works have shown that higher-weight edges are more likely to have that corresponding edge present on the other side of the nervous system (*Gerhard et al., 2017*; *Ohyama et al., 2015*). Here, we provide evidence that considering networks formed from only strong edges also decreases asymmetry at a broad, network-wide level.

To make these two thresholding schemes more comparable, we also examined these results as a function of the proportion of edges from the original network which that threshold removed (*Figure 6E and F*, lower x-axis). We found that when thresholding based on synapse counts, the majority (~60%) of the edges of the networks need to be removed for any test (in this case the density-adjusted group connection test) to yield non-significant p-values. Conversely, for the thresholds based on input proportion, the density-adjusted group connection test yielded a p-value greater than 0.05 after removing only the weakest ~20% of edges. Strikingly, we observed that when considering only the strongest ~60% of edges in terms of input proportion, even the density test had a high p-value (>0.05), while for the synapse-based thresholds we examined, this never occurred. We observed similar trends when running a thresholding experiment in isolation on the KC → KC subgraph (*Figure 6— figure supplement 1*).

These findings are consistent with previous work in connectomics which has hinted at the importance of input proportion as a meaningful 'edge weight.' *Gerhard et al., 2017*, compared the connectivity of select neurons in the nerve cord between L1 and L3 stages of the larva. They observed that while the number of synapses from the mdIV cell type onto various nerve cord local neurons can grow ~3- to 10-fold from L1 to L3, the proportion of that downstream neuron's input stays relatively conserved. Based on this finding, the authors suggested that perhaps the nervous system evolved to keep this parameter constant as the organism develops. An analysis of wiring in the olfactory system of the adult *Drosophila* suggested a similar interpretation after examining a projection neuron cell type with an asymmetric number of neurons on the two sides of the brain (*Tobin et al., 2017*). Here, we provide further evidence based on the entire brain of the *Drosophila* larva that while the left and right hemispheres may appear significantly different when considering all observed connections, the networks formed by only the strongest edges (especially in terms of input proportion) are not significantly different between the hemispheres when viewed through the lens of the models considered in this work.

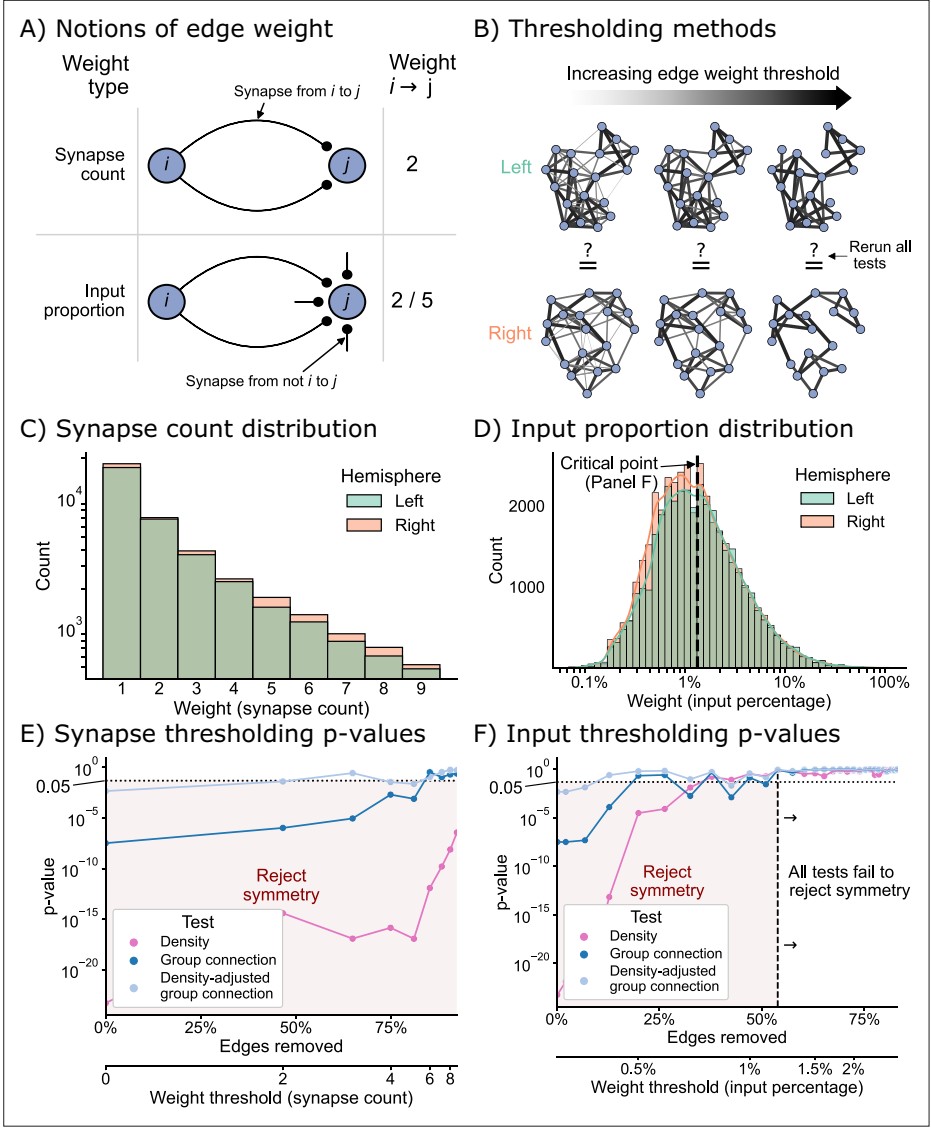

**Figure 6.** The effect of edge weight threshold on the significance level for each of the tests of bilateral symmetry. Diagrams of (**A**) two notions of edge weight and (**B**) application of edge weight thresholds to examine bilateral symmetry. See Edge weight thresholds for more explanation. (**C**) Distribution of synapse count edge weights. The right hemisphere consistently has more edges in each synapse count bin. (**D**) Distribution of input percentage edge weights. The right hemisphere has more edges in the lower (<1%) portion of this distribution, but the hemispheres match well for high edge weights. (**E**) p-Values for each test after synapse count thresholding, plotted as a function of the percentage of edges which are removed from the networks, as well as the corresponding weight threshold (lower x-axis). The p-values for all tests generally increased as a function of synapse count threshold, but the density test never reached a p-value >0.05 over this range of thresholds. (**F**) p-Values for each test after input percentage thresholding, plotted as a function of the percentage of edges which were removed from the networks, as well as the corresponding weight threshold (lower x-axis). Note that all tests yielded insignificant (>0.05) p-values after a threshold of around 1.25% input proportion. Compared to the results in (**E**), thresholding based on input percentage reached insignificant p-values faster as a function of the total amount of edges removed for all tests.

The online version of this article includes the following figure supplement(s) for figure 6:

**Figure supplement 1.** p-Values for left/right comparisons as a function of edge weight thresholds (as in Edge weight thresholds), but performed only on the KC → KC subgraph.

**Table 2.** Summary of tests, models, hypotheses, whether Kenyon cells (KC) were included, and the resulting p-values for each evaluation of bilateral symmetry.

| Test method | Model | $H_0$ (vs $H_A \neq$) | KC | p-Value |
|---|---|---|---|---|
| Density test | ER | $p^{(L)} = p^{(R)}$ | + | $<10^{-23}$ |
| Group connection test | SBM | $B^{(L)} = B^{(R)}$ | + | $<10^{-7}$ |
| Density-adjusted group connection test | DA-SBM | $B^{(L)} = cB^{(R)}$ | + | $<10^{-2}$ |
| Density test | ER | $p^{(L)} = p^{(R)}$ | - | $<10^{-27}$ |
| Group connection test | SBM | $B^{(L)} = B^{(R)}$ | - | $<10^{-2}$ |
| Density-adjusted group connection test | DA-SBM | $B^{(L)} = cB^{(R)}$ | - | ~0.60 |

## Discussion

### Summary

We began with what was at its face a very simple question: is the connectivity on the left and the right side of this brain 'different?' We then described several ways that one could mathematically formalize notions of 'different' from the perspective of network model parameters: difference in density of connections across the entire network (Density test), difference in group connection probabilities (Group connection test), or difference in group connection probabilities while adjusting for a difference in density (Density-adjusted group connection test). We proposed a test procedure corresponding with each of these notions, relying on well-established statistical techniques for evaluating contingency tables and combining p-values to construct our tests. The results of these different test procedures varied markedly (*Table 2*). Specifically, we saw that the network densities were significantly different between the hemispheres. The group connection test also detected a difference, highlighting seven group-to-group connections which had significantly differing connection probabilities when comparing the hemispheres. However, when we added an adjustment to the group connection comparison to account for the difference in network density, this test had only two significant group connections, and both were projections from the Kenyon cells. Thus, the asymmetry observed (at least when viewed through the lens of these high-level network statistics) between the hemispheres can be thought of as a global density difference in addition to a cell type-specific effect shown in the Kenyon cells. We confirmed this finding by simply removing the Kenyon cells, and showing that the density-adjusted group connection test no longer rejected (Removing Kenyon cells). Finally, we examined whether the left and right hemisphere networks would become less dissimilar when only high-edge-weight edges were considered (Edge weight thresholds). We found that whether thresholding based on number of synapses or the proportion of input to the post-synaptic neuron, p-values generally increased for each test (i.e. less significant asymmetry was detected) as the edge weight threshold grew. However, we observed that thresholds based on neuron input proportion could achieve symmetry while removing fewer (only 20% for some tests) edges. These results are consistent with the idea that the nervous system evolved to preserve a relative balance of inputs to individual neurons, which has been suggested by previous studies on specific subcircuits in the larval and adult *Drosophila* nervous system (*Gerhard et al., 2017*; *Tobin et al., 2017*; *Berck et al., 2016*).

### Limitations

As with any statistical inference, our conclusions are valid under particular model assumptions. Therefore, it is important to highlight the assumptions which motivated each of our tests in order to understand what each p-value means (and what it does not). We highlight several of these assumptions below, and comment on alternative assumptions that one could make in each case.

#### What model?

First, while we motivated the tests presented here by assuming that some statistical model produced the connectivity of the left and the right hemispheres, these models do not literally describe the

process which generated these networks. However, without knowledge of how genes and development give rise to the connectome, we know of no more correct model for how this connectome was generated (*Vogelstein et al., 2019*; *Witvliet et al., 2021*; *Barabási and Barabási, 2020*) (and even this would still be just a model). Without an agreed upon definition of bilateral symmetry, we chose to start from the simplest definition of what one *could* mean by bilateral symmetry. From this simplest network model, we iteratively added complexity to the definition of bilateral symmetry until we found the simplest model for which the *Drosophila* larva connectome displayed no significant asymmetry. We also note that previous studies have found associations between the test statistic we study here (graph or subgraph density) and various other biological properties, such as development (*Witvliet et al., 2021*), neurodegeneration (*Pfeiffer et al., 2020*), and phylogenetics (*Suarez et al., 2022*).

However, many other network models could have been applied to examine different definitions of bilateral symmetry. For instance, SBM may fail to capture certain features of an empirically observed network, such as degree distributions. This led to the development of the popular degree-corrected SBM (*Karrer and Newman, 2011*), which adds parameters to account for heterogeneous node degrees. A modified group connection test which also compares these degree correction parameters would be a natural extension of the current work, but requires further study to establish as a valid statistical test. Tests based on the random dot product graph model (*Tang et al., 2017*; *Athreya et al., 2018*; *Chung et al., 2022*) would allow us to compare connection probabilities between hemispheres without assuming that neurons belong to a finite number of groups. *Bravo-Hermsdorff et al., 2021*, showed that a two-network-sample test could be constructed from subgraph counts, which they argue characterize a network's 'texture' rather than its 'backbone' as studied in this work. We also did not use network models that incorporate edge weights, as two-network-sample tests for this case are even less developed than for the unweighted case. Further, a variety of neuroscience-specific network models (such as those which incorporate spatial information) have been proposed (*Váša and Mišić, 2022*). Nevertheless, we note that even if one is concerned with these more elaborate notions of symmetry, they are still related to the simple models studied here. For instance, the network density would affect a network's latent positions under the random dot product graph model, as well as the count of any possible subgraph. Thus, even if one prefers a different definition of bilateral symmetry, the definitions presented here were worth testing.

## What is a cell type?

Second, even if these networks were generated from SBMs, alternative groupings of neurons could have been used. We used broad cell type categorizations from previous literature (*Winding et al., 2023*) to partition our network into groups. However, we could have used a coarser partition, categorizing neurons as sensory, interneuron, and descending/output. Conversely, we could have used a finer partition, splitting the cell types used here into subgroups (such as whether a sensory neuron receives odor or visual information). As these different partitions likely lead to different subgraph sizes and connection probabilities, the statistical power of the group connection test would also be affected by these choices (*Helwegen et al., 2023*). Thus, the results presented for any group connection test need to be interpreted in terms of the specific cell type groupings used.

Further, a rich literature exists on *inferring* the partition for an SBM from the observed connectivity (*Lee and Wilkinson, 2019*; *Peixoto, 2014*; *Peixoto, 2017*; *Rohe et al., 2011*; *Sussman et al., 2012*; *Funke and Becker, 2019*) – this is one perspective for clustering neurons based on their observed connectivity, much like clustering procedures are used to predict meaningful groups of neurons based on morphology, activity, or gene expression. Applying these techniques to a connectome would yield alternative groupings of neurons (as in *Winding et al., 2023*) to use for a group connection test, which again could change its conclusions. However, this approach requires further study, as it introduces a new source of uncertainty since more model parameters are estimated from the data.

## What about neuron pairs?

Third, we assumed that the two networks we observed were *unmatched* – that is, the tests we applied did not use any pairing of individual neurons between hemispheres. In *Drosophila*, this 1-to-1 neuron correspondence is known to exist for most neurons, particularly in the larva. GAL-4 lines are able to reliably label bilateral neuron pairs on the basis of their gene expression (*Jenett et al., 2012*; *Eschbach et al., 2020*). These neurons tend to be similar in terms of their morphology and their

connectivity (*Winding et al., 2023*; *Ohyama et al., 2015*; *Pedigo et al., 2022*; *Schlegel et al., 2021*; *Eschbach et al., 2020*; *Gerhard et al., 2017*; *Schneider-Mizell et al., 2016*). Methods which use this pairing (e.g. *Tang et al., 2017*; *Ghoshdastidar and Von Luxburg, 2018*; *Bhadra et al., 2019*, as well as tests based on correlated ER and SBM models) would be able to evaluate symmetry in light of edge correspondences between the two networks, and could have higher power at detecting certain asymmetries. However, these methods assume that the matching of nodes is *perfect* and *complete* – if even one neuron pairing is a mistake, or if even one neuron does not have a partner in the opposite hemisphere, then these tests could be invalid or inapplicable. We note that graph matching techniques could estimate a correspondence between nodes for all neurons (*Fishkind et al., 2019*; *Vogelstein et al., 2015*; *Saad-Eldin et al., 2021*; *Winding et al., 2023*; *Pedigo et al., 2022*); however, the statistical consequences of first learning this (likely imperfect) alignment prior to using a method which assumes the alignment is known and exact have not been thoroughly studied, so we did not explore it further here.

## Outlook

We presented the first statistical comparison of bilateral networks in a neuron-level brain connectome. While we focused on the larval *Drosophila* brain connectome, these techniques could be applied to future connectomes to evaluate bilateral symmetry in other individuals or organisms. More generally, we presented several notions that can be used to compare two networks, a particularly relevant problem in the current age of connectomics. Human (macroscale) connectomics has seen an explosion in the number of network samples that can be obtained, allowing for different approaches for comparing connectomes across populations, from simple comparisons of edges (*Ingalhalikar et al., 2014*) to low-rank and sparse regressions across networks (*Xia et al., 2020*). However, nanoscale connectomics is still technologically limited in its acquisition rate, often to only one or at best a few (<10, e.g. *Witvliet et al., 2021*) individuals for a given experiment. Nevertheless, we wish to make valid inferences and comparisons between these connectomes (*Vogelstein et al., 2019*; *Barsotti et al., 2021*; *Abbott et al., 2020*; *Galili et al., 2022*). The framework for two-network-sample testing presented here will facilitate these kinds of comparisons. To make these comparisons more practical to neuroscientists, we demonstrated the importance of adjustments to simple null hypotheses – as we saw, even a difference in something as simple as a network density can be related to other network comparisons. For example, take the problem of comparing the connectome of the larval and adult *Drosophila*. Since the adult *Drosophila* brain has orders of magnitude more nodes (*Raji and Potter, 2021*; *Winding et al., 2023*; *Bates et al., 2020*), the density of this network is likely to be smaller than that of the larva. Therefore, we may want to consider a more subtle question – are the connectomes of the adult and larva different (and if so, how) after adjusting for this difference in density? These kinds of biologically motivated adjustments to out-of-the-box statistical hypotheses will be key to drawing valid inferences from connectomes which are also relevant to meaningful questions in neuroscience.

## Methods

### Network construction

Here, we explain how we generated networks for the bilateral symmetry comparison. We started from a network of all neurons in the brain and sensory neurons which project into it for a larval *Drosophila* (*Winding et al., 2023*). As in *Winding et al., 2023*, we removed neurons which were considered partially differentiated. From this network, we selected only the left-to-left (ipsilateral) induced subgraph, and likewise for the right-to-right. We ignored a pair of neurons which had no left/right designation, as their cell bodies lie on the midline (*Winding et al., 2023*). To ensure we had fully connected networks on either hemisphere, we took the largest weakly connected component of neurons on the left, and likewise on the right.

With this selection for our nodes of interest, we then choose our set of edges to be:

- *Unweighted*: we only considered the presence or absence of a connection, creating a binary network. For most analyses except where explicitly indicated, this meant we considered an edge to exist if there was at least one synapse from the source to the target neuron. For this connectome, four edge types are available: axo-axonic, axo-dendritic, dendro-dendritic, and dendro-axonic. We made no distinction between these four edge types when constructing the

binary networks. One could consider notions of bilateral symmetry for a weighted network, but we focused on the unweighted case for simplicity and the fact that most network models are for binary networks. We studied the effect of varying the edge weight requirement (i.e. the threshold) for an edge to exist in Edge weight thresholds.

- *Directed*: we allow for a distinction between edges which go from neuron $i$ (pre-synaptic) to neuron $j$ (post-synaptic) and the reverse.
- Loopless: we remove any edges which go from neuron $i$ to neuron $i$, as the theory on network testing typically makes this assumption. We note that while ~18% of neurons have a connection to themselves, these self-loops comprise only ~0.7% of edges.

When comparing two networks, methods may make differing assumptions about the nature of the two networks being compared. One of the most important is whether the method assumes a correspondence between nodes (*Tantardini et al., 2019*). Some methods (matched comparisons, also called known node correspondence) require that the two networks being compared have the same number of nodes, and that for each node in network 1, there is a known node in network 2 which corresponds to it. Other methods (unmatched comparisons, also called unknown node correspondence) do not have this requirement. To make an analogy to the classical statistical literature on two-sample testing, this distinction is similar to that between an unpaired (unmatched) and a paired (matched) t-test. We focused on the unmatched case in this work, where we say nothing about whether any neurons on the left correspond with any specific neurons on the right.

## Two-network-sample testing

Here, we describe in more detail the methods used to evaluate bilateral symmetry, each of which is based on some generative statistical model for the network. For each model, we formally define the model, describe how its parameters can be estimated from observed data, and then explain the test procedure motivated by the model. A more thorough review of these models can be found in *Chung et al., 2021*.

## Independent edge random networks

Many statistical network models fall under the umbrella of independent edge random networks, sometimes called the inhomogeneous ER model. Under this model, the elements of the network's adjacency matrix $A$ are sampled independently from a Bernoulli distribution:

$$A_{ij} \sim Bernoulli(P_{ij})$$

If $n$ is the number of nodes, the matrix $P$ is an $n \times n$ matrix of probabilities with elements in $[0, 1]$. Depending on how the matrix $P$ is constructed, we can create different models. We next describe several of these choices. Note that for each model, we assume that there are no loops, or in other words the diagonal of the matrix $P$ will always be set to zero.

## ER model and density testing

Perhaps the simplest model of a network is the ER model. This model treats the probability of each potential edge in the network occurring to be the same. In other words, all edges between any two nodes are equally likely. Thus, for all $(i, j), i \neq j$, with $i$ and $j$ both running from $1...n$, the probability of the edge $(i, j)$ occurring is

$$P[A_{ij} = 1] = P_{ij} = p$$

where $p$ is the global connection probability.

Thus, for this model, the only parameter of interest is the global connection probability, $p$. This is sometimes also referred to as the network density. For a directed, loopless network, with $n$ nodes, there are $n(n-1)$ unique potential edges (since we ignore the $n$ elements on the diagonal of the adjacency matrix). If the observed network $A$ has $m$ total edges, then the estimated density is simply

$$\hat{p} = \frac{m}{n(n-1)}.$$

In order to compare two networks $A^{(L)}$ and $A^{(R)}$ under this model, we simply need to compute these estimated network densities ($\hat{p}^{(L)}$ and $\hat{p}^{(R)}$), and then run a statistical test to see if these

densities are significantly different. Under this model, the total number of edges $m$ comes from a $Binomial(n(n-1), p)$ distribution. This is because the number of edges is the sum of independent Bernoulli trials with the same probability. If $m^{(L)}$ is the number of edges on the left hemisphere, and $m^{(R)}$ is the number of edges on the right, then we have:

$$m^{(L)} \sim Binomial(n^{(L)}(n^{(L)} - 1), p^{(L)})$$

and independently,

$$m^{(R)} \sim Binomial(n^{(R)}(n^{(R)} - 1), p^{(R)})$$

To compare the two networks, we are interested in a comparison of $p^{(L)}$ vs. $p^{(R)}$. Formally, we are testing:

$$H_0 : p^{(L)} = p^{(R)}, \quad H_a : p^{(L)} \neq p^{(R)}.$$

Fortunately, the problem of testing for equal proportions under the binomial is well studied. In our case, we used a chi-squared test (*Agresti, 2013*) to run this test for the null and alternative hypotheses above.

## SBMs and group connection testing

An SBM is a popular statistical model of networks (*Holland et al., 1983*). Put simply, this model treats the probability of an edge occurring between node $i$ and node $j$ as purely a function of the communities or groups that node $i$ and $j$ belong to. This model is parameterized by:

- An assignment of each node in the network to a group. Note that this assignment can be considered to be deterministic or random, depending on the specific framing of the model one wants to use. Here, we are assuming $\tau$ is a fixed vector of assignments. We represent this non-random assignment of neuron to group by an $n$-length vector $\tau$. If there are $K$ groups, $\tau$ has elements in $\{1...K\}$. If the $i$-th element of $\tau$ is equal to $k$, then that means that neuron $i$ is assigned to group $k$.
- A set of group-to-group connection probabilities. We represent these probabilities by the matrix $B \in [0,1]^{K \times K}$, where the element $(k, l)$ of this matrix represents the probability of an edge from a neuron in group $k$ to one in group $l$.

Thus, the probability of any specific edge $(i, j)$ can be found by looking up the appropriate element of $B$:

$$P[A_{ij} = 1] = P_{ij} = B_{\tau_i, \tau_j}$$

In our case, we assume $\tau$ is known – in the case where it is not, or one simply wishes to estimate an alternative partition of the network, many methods exist for estimating $\tau$. But with $\tau$ known, estimating $B$ becomes simple, amounting to doing $K^2$ subgraph density estimates. Specifically, let $m(k, l)$ be the number of edges from nodes in group $k$ to nodes in group $l$. We then compute the density of this subgraph for each $(k, l)$ pair (ignoring self-loops):

$$\hat{B}_{k,l} = \begin{cases} \frac{m(k,l)}{n_k n_l}, & \text{if } k \neq l \\ \frac{m(k,l)}{n_k(n_k - 1)}, & \text{if } k = l \end{cases}$$

where $n_k$ is the number of nodes in group $k$, and likewise for $n_l$.

Assuming the SBM, we are interested in comparing the group-to-group connection probability matrices, $B$, for the left and right hemispheres. The null hypothesis of bilateral symmetry becomes

$$H_0 : B^{(L)} = B^{(R)}, \quad H_A : B^{(L)} \neq B^{(R)} \tag{4}$$

Rather than having to compare one proportion as in ER model and density testing, we are now interested in comparing all $K^2$ probabilities between the SBM models for the left and right hemispheres. The hypothesis test above can be decomposed into $K^2$ hypotheses. $B^{(L)}$ and $B^{(R)}$ are both $K \times K$ matrices, where each element $B_{kl}$ represents the probability of a connection from a neuron in group $k$ to one in group $l$. We also know that group $k$ for the left network corresponds with group $k$

for the right. In other words, the *groups* are matched. Thus, we are interested in testing, for $k, l$ both running from $1...K$:

$$H_0 : B_{kl}^{(L)} = B_{kl}^{(R)}, \quad H_A : B_{kl}^{(L)} \neq B_{kl}^{(R)} \tag{5}$$

Now, we are left with $K^2$ p-values from *Equation 5*, each of which bears upon the overall null hypothesis in *Equation 4*. We therefore require some method of combining these p-values into one, or otherwise making a decision about the hypothesis in *Equation 4*. Many methods for combining p-values have been proposed. This problem of combining p-values can itself be viewed as a hypothesis testing problem. Denoting the $(k, l)$ th p-value from *Equation 5* as $p_{kl}$, we are testing

$$H_0 : p_{kl} \sim Uniform(0, 1)$$

versus the alternative hypothesis that at least one of the p-values is distributed according to some non-uniform, non-increasing density with support $[0, 1]$ (*Birnbaum, 1954*; *Heard and Rubin-Delanchy, 2018*). *Birnbaum, 1954*, showed that no method of combining these p-values can be optimal in general to all alternatives, so we are left with a decision to make (with no universally preferred answer) about which methods to use to combine p-values (*Heard and Rubin-Delanchy, 2018*). Here, we select Tippett's method (*Tippett, 1931*; *Heard and Rubin-Delanchy, 2018*) due to its ubiquity, simplicity, and power against various alternatives to bilateral symmetry under a simulation described in Power and validity of group connection test under various alternatives (*Figure 3—figure supplement 4*). In future work, specific classes of alternatives may motivate different methods for combining p-values, as described in *Heard and Rubin-Delanchy, 2018*.

We also examined the p-values from each of the individual tests after Holm-Bonferroni correction to correct for multiple comparisons. As in ER model and density testing, we used chi-squared tests (*Agresti, 2013*) to perform each of the individual hypothesis tests in *Equation 5*. Note also that in some cases, an element of $B^{(L)}$ and/or $B^{(R)}$ could be 0; in each of these cases, we did not run that specific comparison between elements, as the notion of testing for proportions being the same becomes nonsensical. We indicated these tests in *Figure 3C*, *Figure 4C*, and *Figure 5C–D*, and note that these tests were not included when computing the number of comparisons for the Holm-Bonferroni correction. We also note that when few edges (say, <10 are present in a given subgraph), exact tests (e.g. Fisher's exact test; *Agresti, 2013*) may be more appropriate, as they do not rely on asymptotic approximations. We found that in the current work, this choice of test did not substantially affect the results (*Figure 3—figure supplement 5*).

## Density-adjusted group connection testing

In density-adjusted group connection test, we considered the null hypothesis that the left hemisphere connection probabilities under the SBM are a scaled version of those on the right:

$$H_0 : B^{(L)} = cB^{(R)} \text{ vs. } H_A : B^{(L)} \neq cB^{(R)}. \tag{6}$$

The scale for this comparison is the ratio of the densities between the left and the right hemisphere networks:

$$c = \frac{p^{(L)}}{p^{(R)}}. \tag{7}$$

Analogous to the group connection testing in *Equation 5*, this means that the individual group connection hypotheses become

$$H_0 : B_{kl}^{(L)} = cB_{kl}^{(R)}, \quad H_A : B_{kl}^{(L)} \neq cB_{kl}^{(R)}. \tag{8}$$

where $c$ can be viewed as a probability ratio:

$$B_{kl}^{(L)} \overset{?}{=} cB_{kl}^{(R)}$$

$$\frac{B_{kl}^{(L)}}{B_{kl}^{(R)}} \overset{?}{=} c$$

In essence, we wish to test whether this probability ratio for each subgraph matches a prespecified hypothesized value, $c$. To test *Equation 5*, we used a modified score test (*Miettinen and Nurminen, 1985*), which aims to determine whether the ratio of two proportions is significantly different from some known constant, $c$. Note that this test reduces to the standard chi-squared test when the probability ratio $c = 1$. We used this score test in the individual group connection tests, with all other machinery (e.g. for combining p-values or correcting for multiple comparisons) remaining the same as in SBMs and group connection testing. We found that the results using this score test agreed well with an intuitive approach to performing the density adjustment wherein we randomly removed edges from the right hemisphere to set the densities of the networks equal, and then re-ran the standard group connection test over many resamples (*Figure 4—figure supplement 1*). Again, it is worth noting that when testing on very sparse subgraphs, exact versions of this test may be advisable, though these are computationally more difficult to implement (*Chan, 1998*).

### Edge weight thresholds

To examine the effect of which edges are used to define the left and right networks on the p-values from each test, we tested various edge weight thresholds used to define our binary networks for comparison. Given a set of edges (i.e. $(i, j)$ pairs) with corresponding weights $w_{ij}$, a thresholding $\mathcal{E}(t)$ simply selects the subset of those edges for which $w_{ij}$ is greater than or equal to some threshold, $t$.

$$\mathcal{E}(t) = \{(i, j) : w_{ij} \geq t\}$$

Let $s_{ij}$ be the observed number of synapses from neuron $i$ to neuron $j$. We considered two thresholding schemes: the first was to simply use the number of synapses from neuron $i$ to $j$ as the edge weight and the second was to consider the edge weight from neuron $i$ to $j$ to be the number of synapses from $i$ to $j$ divided by the total number of observed synapses onto neuron $j$. We stress that the number of synapses onto neuron $j$ is not necessarily equal to the weighted degree of neuron $j$. This is simply because we consider all annotated post-synaptic contacts onto neuron $j$, and some number of those contacts may not be connected to another neuron in the current networks considered here. We denote the number of synapses onto neuron $j$ as $D_j$. To summarize:

- Synapse number threshold: $w_{ij} = s_{ij}$
- Input proportion threshold: $w_{ij} = \frac{s_{ij}}{D_j}$

Given either definition of the weighting scheme, we formed a series of networks by varying the edge weight threshold, $t$. We stress that edge weights were used *only* for the purposes of defining the edges to consider for our (binary) networks – the edge weights themselves were not used in the statistical tests. We then re-ran the density, group connection, and density-adjusted group connection tests for each network. The p-values for these tests are plotted against the weight thresholds and the proportion of edges removed in *Figure 6E and F* for the synapse number and input proportion thresholds, respectively.

### Power and validity of group connection test under various alternatives

In SBMs and group connection testing, we considered the group connection test, where the goal was to test

$$H_0 : B^{(L)} = B^{(R)} \text{ vs. } H_A : B^{(L)} \neq B^{(R)}. \tag{9}$$

We saw that this set of hypotheses could be decomposed into $K^2$ (where $K$ is the number of groups) different hypotheses

$$H_0 : B_{kl}^{(L)} = B_{kl}^{(R)}, \quad H_A : B_{kl}^{(L)} \neq B_{kl}^{(R)}, \tag{10}$$

yielding a p-value for the $(k, l)$ th test, $p_{kl}$. We now consider the problem of trying to combine these p-values into one which bears on the overall hypotheses in *Equation 9*. We proposed using Tippett's method for combining p-values (*Tippett, 1931*), and we now demonstrate the utility of this method against various alternatives.

To do so, we performed the following simulation experiment. First, we consider two hypothetical group connection matrices, $B^{(1)}$ and $B^{(2)}$. We set $B^{(1)} = \hat{B}^{(L)}$. We also consider the matrix

$M$, which is a $K \times K$ matrix denoting the number of possible edges in each block of an SBM. Here, we again set $M = \hat{M}^{(L)}$, in other words, we use the number of potential edges for each block observed for the left hemisphere network. To analyze the sensitivity of Tippett's method to different alternatives, we conducted the following simulation: Let $t$ be the *number of probabilities to perturb*. Let $\delta$ represent the *strength of the perturbation*. We performed experiments using $\delta \in \{0, 0.1, 0.2, 0.3, 0.4, 0.5\} X t \in \{0, 25, 50, 75, 100, 125\}$ (note that if $\delta = 0$ or $t = 0$, then we are under the null hypothesis in *Equation 9*). For each $(\delta, t)$, we ran 50 replicates of the simulation below:

1. Randomly select $t$ probabilities without replacement from the elements of $B$.
2. For each of the selected elements, set $B_{kl}^{(2)} = TN(B_{kl}^{(1)}, \delta B_{kl}^{(1)})$, where $TN$ is a truncated normal distribution with support $[0, 1]$.
3. For each of the unselected elements, set $B_{kl}^{(2)} = B_{kl}^{(1)}$.
4. For each block $(k, l)$, sample the number of edges in that block for network 1: $m_{kl}^{(1)} \sim Binomial(M_{kl}, B_{kl}^{(1)})$.
   Sample the number of edges in each block similarly for network 2, but using $B^{(2)}$.
5. For each block $(k, l)$, compare $m_{kl}^{(1)}$ and $m_{kl}^{(2)}$ using chi-squared tests as in SBMs and group connection testing. This yields a set of p-values $\mathcal{P} = \{p_{1,1}, p_{1,2} \ldots p_{(K-1),K}, p_{K,K}\}$ for each comparison.
6. Apply Tippett's method to combine the p-values $\mathcal{P}$ into one p-value for the overall hypotheses.

We observed that the p-values obtained from Tippett's method were valid – they controlled the probability of Type I error for any significance level (*Figure 3—figure supplement 4A*). Further, we observed that Tippett's method was also powerful against differing alternatives to the null hypothesis (*Figure 3—figure supplement 4B*). Tippett's method had a power of 1 against the alternative $(t = 25, \delta = 0.5)$, meaning a small number of large perturbations. It also had a power of ~0.8 against the alternative $(t = 125, \delta = 0.1)$, in other words, a large number of small perturbations. Thus, we concluded that Tippett's method is a reasonable choice of method for combining p-values for our group connection test.

## Code and data

Analyses relied on graspologic (*Chung et al., 2019*), NumPy (*Harris et al., 2020*), SciPy (*Virtanen et al., 2020*), Pandas (*McKinney, 2010*), statsmodels (*Seabold and Perktold, 2010*), and NetworkX (*Hagberg et al., 2008*). Plotting was performed using matplotlib (*Hunter, 2007*) and Seaborn (*Waskom, 2021*).

## Acknowledgements

BDP was supported by the NSF Graduate Research Fellowship (Grant no. DGE1746891). JTV was supported by the NSF CAREER Award (Grant no. 1942963). JTV was supported by the NSF NeuroNex Award (Grant no. 2014862). JTV and CEP were supported by the NIH BRAIN Initiative (Grant no. 1RF1MH123233-01). The authors thank members of the NeuroData lab for helpful feedback.

## Additional information

### Funding

| Funder | Grant reference number | Author |
| --- | --- | --- |
| National Science Foundation | DGE1746891 | Benjamin D Pedigo |
| National Science Foundation | 1942963 | Joshua T Vogelstein |
| National Science Foundation | 2014862 | Joshua T Vogelstein |
| National Institutes of Health | 1RF1MH123233-01 | Carey E Priebe Joshua T Vogelstein |

| Funder | Grant reference number | Author |
|--------|------------------------|--------|

The funders had no role in study design, data collection and interpretation, or the decision to submit the work for publication.

## Author contributions

Benjamin D Pedigo, Conceptualization, Data curation, Software, Formal analysis, Supervision, Funding acquisition, Validation, Investigation, Visualization, Methodology, Writing - original draft, Project administration, Writing – review and editing; Mike Powell, Validation, Investigation, Visualization, Methodology, Writing – review and editing; Eric W Bridgeford, Software, Validation, Investigation, Methodology, Writing – review and editing; Michael Winding, Conceptualization, Resources, Data curation, Investigation, Writing – review and editing; Carey E Priebe, Conceptualization, Supervision, Funding acquisition, Methodology, Project administration, Writing – review and editing; Joshua T Vogelstein, Conceptualization, Resources, Supervision, Funding acquisition, Project administration, Writing – review and editing

## Author ORCIDs

Benjamin D Pedigo 
Mike Powell 
Eric W Bridgeford 
Michael Winding 
Carey E Priebe 
Joshua T Vogelstein 

## Decision letter and Author response

Decision letter https://doi.org/10.7554/eLife.83739.sa1
Author response https://doi.org/10.7554/eLife.83739.sa2

## Additional files

### Supplementary files

• MDAR checklist

### Data availability

The code to perform all analyses in this paper (Python 3) can be found at https://github.com/neurodata/bilateral-connectome (MIT license) and viewed as a JupyterBook (*Executable Books Community, 2020*) at http://docs.neurodata.io/bilateral-connectome. The version for this submission is archived at https://doi.org/10.5281/zenodo.7733481 (*Pedigo, 2023*). All data analyzed in this study were generated in *Winding et al., 2023*. These data are also included for convenience in the code repository linked above and as *Figure 1—source data 1*.

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
