## [Editor Report]

This important work demonstrates a significant asymmetry between the connectivity statistics of the left and right hemispheres of the *Drosophila* larva brain. The evidence supporting the conclusions is compelling and represents a first step toward the development of statistical tests for comparing pairs of connectomes more generally. This work will therefore be of interest to the broad neuroscience community.

---

## [Decision Letter]

**Decision letter after peer review:**

Thank you for submitting your article "Generative network modeling reveals quantitative definitions of bilateral symmetry exhibited by a whole insect brain connectome" for consideration by *eLife*. Your article has been reviewed by 2 peer reviewers, and the evaluation has been overseen by a Reviewing Editor and K VijayRaghavan as the Senior Editor. The reviewers have opted to remain anonymous.

The reviewers have discussed their reviews with one another. Both reviewers are very enthusiastic about the paper and have minor comments that should be easy to address (see below).

*Reviewer #1 (Recommendations for the authors):*

This study is very clear and well-presented both at the level of the writing and the figures. I only have a couple of questions/recommendations for the authors:

1) In the stochastic block model approach, the authors find differences in 6 group-to-group connections (that all showed higher probabilities in the right hemisphere consistent with the higher density of the right hemisphere). After adjusting for density, they found two group-to-group connections that remain different. When doing this, their assumption is that the density is uniform across the brain hemisphere. However, it is also possible that the density of the network varies to some degree depending on brain areas and neuron types. Could the authors compute the density for subnetworks or groups of neurons within the hemispheres to determine whether the density is relatively constant within a given hemisphere? For example, removing KC slightly decreased the densities in both hemispheres and slightly increased the difference in densities between the left and the right hemispheres (0,91 versus 0.93 ratios). Having a subnetwork with different trends in density (no difference between left and right or higher density on the left) could potentially introduce a bias when comparing group-to-group probabilities adjusted by density.

2) When exploring the definitions of an edge the authors found that using a threshold to only compare stronger connections and also when using input percentage rather than synaptic count, they were less likely to find differences between the compared network. Did the authors try to compare the connectivity involving KC (they found were different between the hemispheres using SBM) using these edge definitions and thresholds? Do they still find differences?

*Reviewer #2 (Recommendations for the authors):*

My suggestions mainly involve justifying and simplifying assumptions of the approach and discussing how the results could be generalized beyond simple tests of connection density.

Regarding the Erdos-Renyi independent edge weight assumption, I would be curious to see whether the in- and out-degree distributions of the network models (e.g. the SBM) the authors construct are consistent with the empirical data, and if not, a discussion of alternative network models that can match this feature of the data better.

Regarding the observation of KC connectivity being significantly different from the SBM, could this be due to a sample size issue? The inclusion of the number of neurons belonging to each group and/or edges belonging to each comparison would be valuable, and it would also be helpful if the authors could discuss how to deal with issues of statistical power within the SBM.

Finally, I would be interested if the authors could motivate and comment on their focus on connection density as the measure of bilateral symmetry. Certainly, many graph properties may differ between connectomes, connection density being a straightforward one to quantify. It's not clear that the authors' motivating example of comparing the connectome of an organism that has undergone a learning procedure and one that hasn't would be best served by a test of connection density versus something else (like the number of reciprocal connections, degree of convergence, etc.).

---

## [Author Response]

Reviewer #1 (Recommendations for the authors):This study is very clear and well-presented both at the level of the writing and the figures. I only have a couple of questions/recommendations for the authors:1) In the stochastic block model approach, the authors find differences in 6 group-to-group connections (that all showed higher probabilities in the right hemisphere consistent with the higher density of the right hemisphere). After adjusting for density, they found two group-to-group connections that remain different. When doing this, their assumption is that the density is uniform across the brain hemisphere. However, it is also possible that the density of the network varies to some degree depending on brain areas and neuron types. Could the authors compute the density for subnetworks or groups of neurons within the hemispheres to determine whether the density is relatively constant within a given hemisphere? For example, removing KC slightly decreased the densities in both hemispheres and slightly increased the difference in densities between the left and the right hemispheres (0,91 versus 0.93 ratios). Having a subnetwork with different trends in density (no difference between left and right or higher density on the left) could potentially introduce a bias when comparing group-to-group probabilities adjusted by density.

The stochastic block model (and even the density-adjusted version we discuss) does in fact allow for non-uniform density in each brain hemisphere. See, for instance, Figure 3B, which plots the density in each subgraph (where subgraph is a particular set of connections from one group of neurons to another). As the reviewer points out, the density does indeed vary substantially between subgraphs. We apologize for any confusion on this point.

The density-adjustment we present for the SBM comparison is checking whether density *differences* in each subgraph (which we detected using the vanilla SBM test) are consistent with a constant scaling of all connection probabilities on one hemisphere by some constant. We agree that it is possible for some subgraphs to have an opposite trend (e.g., no difference between left and right or higher density on the left as you point out), but this is exactly what our test was designed to determine. As we saw, this constant down-scaling of all right hemisphere subgraph densities was sufficient to make many of the detected differences go away, but not all, indicating that some of these differences you suggest are present.

Finally, we note that if density was adjusted between left/right for each subgraph individually, then all left/right subgraph densities would be the same, leaving us with a nonsensical test.

We have added text to the results where this method is introduced to clarify this point.

“Note that these adjusted hypothesis do not test whether the density across all subgraphs of the left or right hemisphere networks are the same; rather, they are asking wither a single scaling factor (c in Equation 3) makes any significant density *differences* disappear from our previous comparison.”

2) When exploring the definitions of an edge the authors found that using a threshold to only compare stronger connections and also when using input percentage rather than synaptic count, they were less likely to find differences between the compared network. Did the authors try to compare the connectivity involving KC (they found were different between the hemispheres using SBM) using these edge definitions and thresholds? Do they still find differences?

We thank the reviewer for this interesting suggestion for follow-up. We have added Figure 6 —figure supplement 1 which concerns this question. We performed a modified version of the experiment in Figure 6, but only on the KC -> KC subgraph. We observed qualitatively similar trends wherein p-values generally rose for higher values of both types of threshold, but the effect was more pronounced for the input proportion thresholding.

Reviewer #2 (Recommendations for the authors):My suggestions mainly involve justifying and simplifying assumptions of the approach and discussing how the results could be generalized beyond simple tests of connection density.Regarding the Erdos-Renyi independent edge weight assumption, I would be curious to see whether the in- and out-degree distributions of the network models (e.g. the SBM) the authors construct are consistent with the empirical data, and if not, a discussion of alternative network models that can match this feature of the data better.

We thank the reviewer for pointing out this issue of which model to use. Each model will capture (or fail to capture) different aspects of the networks being compared. For instance, as the reviewer suggests, the SBM may fail to capture the node degree distributions completely. We observed this in our data in the comparison (Author response image 1) of the degrees for 100 sampled stochastic block models fit to the left hemisphere network, compared to the observed left hemisphere network itself:

**Author response image 1. sa2fig1:** 

To highlight this point, we added the following text to the discussion:“For instance, stochastic block models may fail to capture certain features of an empirically observed network, such as degree distributions. This led to the development of the popular degree-corrected stochastic block model (Karrer and Newman, 2011), which adds parameters to account for heterogeneous node degrees. A modified group connection test which also compares these degree correction parameters would be a natural extension of the current work, but requires further study to establish as a valid statistical test.”

We fully support (and are ourselves interested in) the study of tests which also include these degree-correction parameters, and a comparison thereof.

However, we also point out that the distribution of the network statistics our tests are based on (e.g. the density or subgraph density) should be relatively preserved under these alternative models. For example, comparing the distribution of densities from 1000 Erdos-Renyi (ER) and degree-corrected Erdos-Renyi (DCER) models (both fit to the left hemisphere network), we see they are quite similar (Author response image 2):

This suggests that our test would still properly control type-I error even if the true underlying network distribution were a DCER model. Since each subgraph in a degree-corrected SBM is essentially one of these DCER graphs, we also expect this would hold in that case. Further study would be required to confirm whether this is holds generally, so we refrain from claiming anything this strong in the manuscript.

Regarding the observation of KC connectivity being significantly different from the SBM, could this be due to a sample size issue? The inclusion of the number of neurons belonging to each group and/or edges belonging to each comparison would be valuable, and it would also be helpful if the authors could discuss how to deal with issues of statistical power within the SBM.

We thank the reviewer for pointing out this question of statistical power for further clarification. We agree that it is important to consider the power of each comparison, and in particular, that a lack of a detected difference can be due to no difference being present (i.e. the null hypothesis being true) OR a lack of statistical power. We took this opportunity to clarify by adding Figure 3 —figure supplements 2 and 3.

Figure 3 —figure supplement 2 (repeated below) shows empirical power of the test we are using to compare subgraphs for a range of number of neurons (and thus, number of potential edges) as well as a range of connection probabilities. We hope this will help readers calibrate their expectations as to how much power they can expect this test to have in future applications, and provide an intuition for how power will change with these two variables.

Figure 3 —figure supplement 3 shows a similar empirical power calculation in simulation, but for the specific numbers of nodes and estimated connection probabilities for each subgraph compared in our application.

As the reviewer pointed out, the test appears to have high power in the simulations based on several of the Kenyon cell-related subgraphs (e.g. KC -> KC). We also added the following text to the section on the stochastic block model to stress this point:

“We stress that, as with any statistical test, a lack of a significant difference (e.g. in other subgraphs) could be the result of the null hypothesis of no difference being true, or simply from a lack of power against a particular alternative (see Figure 3 —figure supplement 2 and Figure 3 —figure supplement 3 for analysis of the power of this test in simulation, and Helwegen et al. (2023) for an excellent discussion on this point).”

In terms of improving the power of the current test, we note that we are already using a standard test for this setting of comparing binomials. For the right application, an improvement in power would likely come from utilizing matched versions of these test (e.g. McNemar’s test) if neuron/edge pairs are known, as we describe in the “What about neuron pairs?” section of the discussion. Finally, we note that collecting multiple connectome samples from each treatment/control group (and using the corresponding generalizations of our (or related) tests to more than 2 connectomes) may be necessary to detect more subtle differences.

Finally, I would be interested if the authors could motivate and comment on their focus on connection density as the measure of bilateral symmetry. Certainly, many graph properties may differ between connectomes, connection density being a straightforward one to quantify. It's not clear that the authors' motivating example of comparing the connectome of an organism that has undergone a learning procedure and one that hasn't would be best served by a test of connection density versus something else (like the number of reciprocal connections, degree of convergence, etc.).

We thank the reviewer for pointing out this issue for further clarification, as we agree it is an important one.

It is true that, a priori, one may not know exactly which changes in graph properties will be related to a particular phenotype. If a specific network statistic is known to vary with some phenotype ahead of time, then it is likely that a more specific test could have more power to detect that change. We support the development of other network tests as the need arises in specific applications, but we note that further work will be required to establish valid statistical approaches to testing in these other settings. We chose to base our tests on one of the simplest network statistics to start with, because it enabled us to generate a principled test wherein the parameter of an actual generative model could be compared between samples.

We also note that many other network features are highly related to network density. See for example the characterization of the correlation of network density with many other graph features (over the complete set of small networks with 9 nodes, in this case) from Chen, Hang, et al. IEEE transactions on visualization and computer graphics (2019):

For an example of this effect in practice in neuroscience, see for example Figure 3—figure supplement 5 in Suarez et al. eLife (2022), which shows how correlated the network density is with many other network features (e.g. transitivity, betweeness, clustering coefficient, etc.).

Finally, we note that several previous studies have noted changes in connection density in particular subgraphs between samples which vary in some other feature. For instance, Witvliet et al. Nature (2021) noticed changes in connection density across development in the *C. elegans* worm. Pfeiffer et al. Experimental Eye Research (2020) saw the addition of new connections between particular cell types in a rabbit model of retinitis pigmentosa. We added a sentence in the discussion to this effect:

“We also note that previous studies have found associations between the test statistic we study here (graph or subgraph density) and various other biological properties, such as development (Witvliet et al., 2021), neurodegeneration (Pfeiffer et al., 2020), and phylogenetics (Suárez et al., 2022).”